# Molecular Investigations of Two First *Brucella suis* Biovar 2 Infections Cases in French Dogs

**DOI:** 10.3390/pathogens12060792

**Published:** 2023-06-01

**Authors:** Guillaume Girault, Vitomir Djokic, Fathia Petot-Bottin, Ludivine Perrot, Bourgoin Thibaut, Hoffmann Sébastien, Acacia Ferreira Vicente, Claire Ponsart, Luca Freddi

**Affiliations:** 1EU/WOAH & National Reference Laboratory for Brucellosis, Animal Health Laboratory, Anses/Paris-Est University, 94700 Maisons-Alfort, France; 2Clinique Vétérinaire de la Ville Haute, 40400 Tartas, France; 3Clinique Vétérinaire des Genêts, 30300 Beaucaire, France

**Keywords:** *Brucella suis* biovar 2, molecular epidemiology, dog infection, wildlife, molecular typing

## Abstract

Despite *Brucella suis* biovar 2’s (BSB2) active circulation in wildlife, no canine infections have been reported. The present paper is the first to describe two cases of BSB2 infections in French dogs. The first case occurred in 2020 and concerned a 13-year-old male neutered Border Collie with clinical signs of prostatitis. The urine culture revealed the excretion of significant levels of *Brucella* in the sample. The second case concerned a German Shepherd with bilateral orchitis, in which it was possible to detect *Brucella* colonies following neutering. HRM-PCR and classical biotyping methods classified both isolated strains as BSB2, in contrast to expected *B. canis*, which is usually the etiological agent of canine brucellosis in Europe. The wgSNP and MLVA analyses highlighted the genetic proximity of two isolates to BSB2 strains originating from wildlife. No pig farms were present in the proximity of either dog’s residence, ruling out potential spill over from infected pigs. Nevertheless, the dogs used to take walks in the surrounding forests, where contact with wildlife (i.e., wild boars or hares, or their excrements) was possible. These cases highlight the importance of adopting a One Health approach to control the presence of zoonotic bacteria in wild animals and avoid spillovers into domestic animals and, potentially, humans.

## 1. Introduction

Brucellosis is a worldwide zoonotic disease caused by *Brucella* spp. Their pathobiology in mammalians has been extensively studied [1,2] due to the public health importance of minimizing infections [3,4]. In France, like in other EU countries, the prevention, control and eradication of brucellosis is regulated by Animal Health Law (AHL; regulation EU No 2018/1882; EUR-Lex-32018R1882-EN-EUR-Lex (europa.eu)). The AHL includes the surveillance and management of *B. abortus*, *B. melitensis* and *B. suis* in production animals and wildlife. However, the AHL only considers dogs as potential reservoirs for *B. abortus*, *B. melitensis* and *B. suis.* In France, surveillance is organised through departmental veterinary diagnostic laboratories, which perform first-line analyses. When the suspected *Brucella* colonies are detected, the isolated strains are sent to National Referent Laboratory (LNR) for animal brucellosis, genetic confirmation and further species characterisation.

Swine brucellosis caused by *Brucella suis* biovar 2 (BSB2) re-emerged in France in the 1990s following the development of free-ranged (outdoor) farms, causing major economic losses to the industry [5]. Domestic pigs acquire *B. suis* through contact with wildlife reservoirs (hares and wild boars) [6]. Occasionally, infection can be imported into intensive farms through contaminated breeding animals (boars) and/or their semen [7]. From 1993 to 2014, the number of outbreaks in France ranged from 0 to 12 annually, with a total of 94 cases reported in this period [8]. Although outbreaks principally occurred in the western part of the country, where pig breeding and, more specifically, outdoor farms are the most concentrated, several incidents were also reported in the south-east after 2012. One epidemiological study of BSB2 causing brucellosis in French pork production identified that the likelihood of transmission of infection among free-range farms is low due to lack of commercial exchange in animals [5], suggesting random contacts with infected wildlife as a major source of infection. The BSB2 is widely present in the wild boar population, which has increased enormously since the 1990s due to a restricted hunting [9,10]. Since 1993, between 20 and 35% of positive *Brucella* serological reactions (determined via the Rose Bengal Test and/or the Complement Fixation Test) were observed in wild boars that were hunted or found dead in several departments throughout the country [5]. However, the proportion of positive serological reactions in the tests carried out on commercially reared pigs in quarantine stations and semen collection centers dropped from 4% in 2012 to 0.6% in 2014 [8]. Hares are an equally important reservoir for BSB2, with 28 strains isolated between 1980s and 2000s, although no investigation of the prevalence of infection in this species has yet been conducted. Due to the increasing number of outbreaks and differentiation of livestock-breeding practices, a study commissioned by French authorities sought to assess the risks of BSB2 transmission to humans exposed to naturally infected pigs [11]. This study found that none of the breeders, nor their families, reported brucellosis cases or related symptoms, suggesting low pathogenicity of BSB2 to humans. Seven human cases of brucellosis caused by BSB2 were reported in the country between 2004 and 2016, all of whom had direct contact with wild boars while hunting or preparing their meat for consumption, highlighting how this species might be an emerging pathogen in people with exposure to contaminated carcasses and specific immune statuses [12]. Outside of Europe, *B. suis* biovars 1, 3 and 5 are identified as more pathogenic to humans, as well as other mammals, compared to biovar 2; these biovars actively circulate among feral and domestic animals [13,14,15,16].

Recently, canine brucellosis became primarily associated with *B. canis*. In Western Europe, infections occurred only sporadically with *B. melitensis*, *B. abortus* or *B. suis* in dogs in close contact with infected livestock or wild animals [17,18,19]. Dogs can be infected through consumption of, contact with or inhalation of contaminated secretions, raw milk, aborted fetuses or placentas [17,20,21,22]. Additionally, consumption of contaminated meat is a possible source of canine outbreaks [23]. To date, infections in dogs with *Brucella* spp. have been detected in different parts of North and South America, as well as some European and Asian countries [18,19,23,24,25,26,27,28,29]. In particular, canine infections with *B. suis* following exposure to feral pigs through hunting or contact with their excrements have been reported worldwide [22,29,30]. A cross-sectional study in eastern Australian dogs documented how *B. suis* exposure is relatively common in our canine companions (seroprevalence 6.6%) if they have contact with feral pigs [31]. To date, all reported cases of infections in dogs in which *B. suis* biovar analysis was carried out concern only biovars 1 and 5 [21,23,28,30,32]. However, no study has systematically analysed the circulation of *B. suis* infection in dogs (including serological surveys in European countries), suggesting that canine brucellosis due to this *Brucella* species is potentially under-reported, probably due to the limitations of accurate diagnosis, the lack of regulation and the low probability of human infection [24,33]. Our results demonstrate, for the first time, BSB2 infection of exposed dogs, highlighting the high survival rate of BSB2 in the environment, the potential for infection in canids and how even domestic pets without evident contact with wild animals or pig industries could be sources for human infections.

## 2. Materials and Methods

### 2.1. Bacterial Cultivation and Strains Isolation

All samples collected for bacteriological analysis were analyzed using routine bacteriological procedures in the local veterinary diagnostic laboratories, in accordance with the safety regulation in force. In brief, the urine samples collected from the 13-year-old Border Collie (first reported case) were directly cultivated using Chocolate PolyViteX (BioMerieux, Craponne, France) and chromogenic CPS media (BioMerieux, Craponne, France). In contrast, tissue biopsies of the scrotal region collected from the 5-year-old German Shepherd (second reported case) were cut with a surgical single-use scalpel, and 10 g was homogenised and diluted in 1/2 to 1/5 ratios in phosphate-buffered saline (PBS) solution (0.9%). The homogenate was then plated onto Chocolate PolyViteX and Columbia agar (Sigma-Aldrich, Saint-Quentin-Fallavier, France) with sheep’s blood (5%) media. The resulting colonies of the two cases were purified via replating and analyzed via MALDI-TOF for first-line identification purposes. The two isolates (20-02069-2828 and 22-03912-5948) suspected to be *Brucella* were transferred to the French National Reference Laboratory for Animal Brucellosis (ANSES, Maisons-Alfort, France) to confirm the *Brucella* genus and determine the species, in accordance with the safety regulations in force.

### 2.2. Phenotypic Identification and Characterisation

Isolates were characterized using standard procedures, in accordance with World Organisation for Animal Health (WOAH) guidelines, in BSL-3 facilities [34]. The strains were classically biotyped based on colonial morphology, gram staining, CO_2_ requirement, H_2_S production, oxidase and urease activity, growth on dyes (basic fuchsin and thionin), lysis by phages (Tb, Wb, Iz, R/C) and agglutination with monospecific sera (anti-A, anti-M and anti-R).

### 2.3. Molecular Analyses

Genomic DNA was extracted using the commercially available QIAGEN QIAamp DNA minikit (QIAGEN, Germany) following the manufacturer’s instructions. The Real-Time PCR (RT-PCR) [35], Multiple Locus Variable-number Tandem Repeat Analysis (MLVA)-16 [36] and High-Resolution Melting (HRM) PCR [37] analyses were performed as previously described. The individual DNA samples of 2 isolates were typed with the MLVA-16 panel single-plex PCR and agarose gel method. Clustering and congruence analyses were conducted with BioNumerics 7.6.3 (BioMérieux), using data as a character dataset via the categorical distance coefficient and MST (Minimum Spanning Tree) method. A total of 561 MLVA-16 profiles originating from different countries (29 Belgium, 1 Bulgaria, 4 Croatia, 2 Czech Republic, 9 Denmark, 214 France, 36 Germany, 69 Hungary, 10 Italy, 2 Poland, 96 Portugal, 77 Spain, 2 Switzerland, 5 United Kingdom and 5 not reported) and hosts (8 bovine, 1 caprine, 2 dog, 92 hare, 5 human, 4 ovine, 212 swine and 227 wild boar) available in MLVA database of our strain collection (https://microbesgenotyping.i2bc.paris-saclay.fr/ (accessed on 1 September 2022)) were used in the analysis (Appendix A).

The whole genome sequencing (WGS) of isolated strains was performed using Illumina Nextera XT kit (Illumina). The sequencing run was performed on MiSeq equipment (Illumina) at the Identypath core facility (ANSES, Maisons-Alfort). A *de novo* assembly was performed using Spades 3.11 [38]. In total, 37 and 49 contigs larger than 1000 bp were obtained for isolates 22-3912-5948 (second case isolate) and 20-02069-2828 (first case isolate), with total genome sizes of 3,303,001 and 3,302,055 bp, respectively. The whole genome SNP (wgSNP) analysis was performed using BioNumerics version 7.6.3 (BioMérieux) to trace back the source of infection. The genome of *B. melitensis* strain 16M was used as the reference genome for all the analyses. A total of 139 available *B. suis* genome sequences, originating from different continents (2 Africa, 9 Asia, 100 Europe, 2 Eurasia, 10 North America and 2 South America and 14 not reported) and various countries (2 Argentina, 3 Belgium, 8 China, 7 Croatia, 1 Czech Republic, 2 Denmark, 29 France, 5 Germany, 1 India, 18 Italy, 1 Netherlands, 3 Portugal, 2 Russia, 1 Romania, 4 Slovenia, 21 Spain, 3 Switzerland, 2 United Kingdom, 10 United States, 2 Zimbabwe), were used in this study for comparison purposes (Appendix A). Chimeric genomes of chromosomes 1 and 2 were generated to compare complete and draft genomes [39]. A minimum set of position filters were applied on the SNP matrix: (i) contiguous SNPs were removed (if found in a 10 bp-window), (ii) with non-informative SNPs, (ii) a required minimum of 15-fold coverage for each SNP, and (iv) ambiguous (i.e., non-ACGT bases) and unreliable bases (i.e., Ns) were discarded. The refined SNP matrix was used to generate a maximum parsimony tree using a maximum parsimony algorithm, allowing phylogenetic analyses.

## 3. Results

### 3.1. Presentation of Cases

In April 2020, a 13-year-old male neutered Border Collie presented with thickening of the bladder, as well as urethral duct and prostate pain related to prostatic palpation, was brought to a veterinary clinic. Urine was cloudy, and all other clinical signs were associated with cystitis and/or prostatitis, suggesting an infection of the urogenital tract. The animal presented no evident signs of brucellosis. This dog was acquired through e-commerce at the age of 4 years and originated from South West France (Pyrennées Atlantiques department). Since then, the dog lived in the Landes department (also South West) as the only pet in its household, and was regularly followed by a veterinarian. The dog used to take long walks in the surrounding forests, where it could have been in contact with other domestic and/or wild animals and their excrements. In order to exclude bacterial infection and/or prostatic cancer, the dog was examined in the local veterinary clinic, where routine cytological and bacteriological analyses of urine were performed. Based on initial anamnesis, brucellosis did not fit the differential diagnosis. Therefore, no serological analysis was advised to detect the potential presence of specific antibodies against *Brucella* sp. since the beginning of the infection, thus preventing serological follow-up of the animal. The ultrasound analysis did not show any evidence of tumoral tissues. The cytological results highlighted high levels of leucocytes, confirming an ongoing infection of the urogenital tract; otherwise, negative results were obtained from urine bacteriology. Considering the persistence of clinical signs, a new urine sample was obtained 7 days later, together with ultrasound. The urine culture identified colonies potentially belonging to *Brucella* genus with an amount of 10^4^ colony forming units (CFU) after three days of incubation, confirming the ongoing infection. The dog was treated with Doxycycline for 20 days, without recovery, which led to decision to put it to sleep, considering the dog’s age and deteriorated health condition.

Another case of an infected dog occurred in February 2022. The 5-years-old male German Shepherd presented high fever with an abnormal enlargement of the genital apparatus, suggesting a bilateral orchitis. The dog was born in the Gard department (South of France), where it was adopted by a family living in the countryside of Beaucaire, an area where wild boars are prevalent. The dog was a household pet and was never used for hunting or had contact with animal breeding facilities. As first complementary investigations, a blood analysis was performed, showing leucocytosis and lymphopenia, indicating an ongoing infection. At the local veterinary diagnostic laboratory, indirect serology was performed that targeted only rough *Brucella* sp., with a negative result. Since the dog had no history of brucellosis, the local veterinary clinic initially decided to treat it with a combination of amoxicillin, clavulanic acid and a non-steroidal anti-inflammatory drug. As there was no improvement, a deterioration of the reproductive organs was instead detected via echography. A neutering with scrotectomy was performed to avoid any potential risk of spread and to identify the cause of the infection. Following surgery, treatment was strengthened with the addition of enrofloxacin for 7 days as a second line-antibiotic, after which dog completely recovered. The retrieved testicles were sent to a diagnostic laboratory to perform a general bacterial culture, the result of which showed the presence of colonies suspected to belong to the *Brucella* genus. After the clinical signs retracted, no additional sampling was performed to follow up on the presence of specific BSB2 antibodies, and no shedding of live bacteria occurred.

In both presented cases, no livestock farms were present in the proximity of their residences, ruling out potential contamination from infected domestic animals.

### 3.2. Genomic and Bacteriological Identification

To confirm the *Brucella* genus identification and determine the species, both isolated strains were transferred to the French National Reference Laboratory for Brucellosis (ANSES, Maisons-Alfort). After total DNA extraction, the real-time PCR analysis targeting *IS711*, *bcsp31* and *per Brucella* genes was performed. From both strains, all three targeted genes were amplified, yielding a positive signal. The DNAs were subsequently tested for rapid identification and differentiation of *Brucella* genus using High-Resolution Melting (HRM) PCR assay. Surprisingly, the melting curve profiles matched with classical BSB2 species, instead of *B. canis*, which was the first expected diagnosis, regarding the host species. The standard bacteriological phenotypic identification of isolates (code numbers 20-02069-2828 and 22-03912-5948, respectively) confirmed the *Brucella* genus and the biotyping traits were consistent with the BSB2 (Table 1).

In order to identify the strains and find the origins of infection, phylogenetic investigations were performed. The MLVA analysis confirmed the BSB2 identity of two isolates, and a strong proximity with strains isolated in French suids and human cases was determined (Figure 1). The two isolated strains were clustered only with French isolates. Interestingly, the first strain (20-02069-2828) was grouped into a branch that mainly contains isolates from pigs and wild boars, while the second isolate (22-03912-5948) is grouped into a subgroup that mostly includes strains isolated from cattle and humans.

The wgSNP analysis was conducted on available strains with *B. suis* species (140 isolates for the comparison, in addition to single *B. canis* and *B. melitensis* isolates used as an outlier group) from various geographical regions of the world (Africa, Asia, Europe, North America and South America). A phylogenetic tree using the maximum parsimony algorithm was generated from the SNP matrix, yielding 8849 SNPs after filtering (Figure 2). The two strains were clustered with all BSB2 isolates, which were available from public databases. The two analyzed strains are classified within a subclade from France, Spain and Belgium. Finally, to better correlate the two isolates from dogs compared to BSB2 strains circulating in French pigs and wildlife, a wgSNP analysis of 33 sequences available in our laboratory (returning 9500 SNPs after filtering) highlighted the presence of a subclade encompassing another 15 strains isolated from pigs, wild boars and hares (Figure 3).

## 4. Discussion

Brucellosis is a highly contagious bacterial zoonosis mainly caused by ingestion of contaminated unpasteurized milk, dairy produce or insufficiently thermically treated meat or close contact with secretions of infected animals (semen, aborted or parturition tissues and fluids). Etiological agents of brucellosis are bacteria from *Brucella* genus, with 12 species currently described. Considering that the whole genus shares ~97% genetic similarity [40], it is unclear how host preferences are developed. However, the plasticity of the genus continues to be discovered. In 2017, for example, European amphibians were found to be infected by one such recently identified species (*B. microti*) [41,42], which was already reported in rodents, foxes and wild boars, thus confirming the broad host range of emerging *Brucella* sp. [43]. However, no *Brucella* sp. are characterised by the development of specific clinical symptoms, either in humans or in animal hosts. At the same time, the emergence of new reservoirs for *Brucella* sp. could be linked to a new consumption pattern, especially practices of organic and raw products, higher globalisation of food or the creation of produce markets with consecutively augmented importation rates, as well as increased animal movement. Therefore, the host range of the whole genus continues to be discovered.

In France, the 13- and 5-years old dogs were confirmed as brucellosis infected in 2020 and 2022, respectively, following different persistent symptoms and the detection of positive bacterial culture for *Brucella*. Reported clinical signs (prostatitis, orchitis) were different between the two cases, though they were compatible with brucellosis infection. First-line treatments related to diagnostic hypotheses were not oriented against *Brucella* infections, leading to a delayed diagnosis of brucellosis and misuse of antibiotics. The lack of brucellosis awareness can probably be explained by the fact that France, like many European countries, is free of brucellosis in ruminants and is not considered as an endemic location for canine brucellosis. Therefore, vets do not consider *Brucella* spp. in differential diagnosis of clinically ill dogs.

The preferred hosts of BSB2 strains are domestic and wild pigs, as well as hares and other wildlife; however, it had never been isolated from either domestic nor wild canids, compared to serotypes 1 and 5 [21,23,28,30,32]. Continuous monitoring and genetic characterisation of newly isolated strains in France shows the emergence of two distinct clades with the preferences either for pigs or hares. At the same time, canine brucellosis in Europe was only reported sporadically, being mainly related to *B. melitensis*, *B. abortus* and, more recently, *B. canis.* Canine brucellosis caused by *B. canis* causes miscarriages, infertility, orchitis, epididymitis, endocarditis, uveitis and discospondylitis. Interestingly, *B. canis* is yet to be isolated from pigs, wild boar, hares or other wildlife; therefore, dogs can be considered as the preference host species. At the same time, in the last decade, there was an increase in the detection of *B. suis* in dogs, particularly in Australia and the United States. A recent seroprevalence study, for example, found that nearly one in 10 (9.3%) dogs in eastern Australia, being exposed to feral pigs or their meat products, are seropositive to smooth *Brucella* spp. (including *B. suis*) [44,45,46]. In Australia, *B. suis* biovar 1 is responsible for canine infections. Humans and dogs being in contact with body fluids and/or tissues from infected feral pigs, through hunting, butchering or consumption of uncooked pork, is considered as the most common risk factor for infection [45]. Surprisingly, the two described dogs were found to be infected with BSB2, a species usually isolated from pigs, hares and wild boars [34]. The BSB2 infection of dogs was not previously documented. In France, BSB2 is mainly encountered in hares and wild boars (*Sus scrofa*), and is known to have very low pathogenicity to humans, with only seven human cases confirmed in a period spanning of 12 years (between 2004 and 2016). All patients had direct contact with wild boars while hunting or preparing wild boar meat [12]. The absence of BSB2-reported cases in dogs may be related to the low level of pathogenicity, as well as potential asymptomatic or latent infections in dogs. It may be relevant to highlight that only the German Shepherd dog was tested for *B. canis* (rough species) antibodies, excluding the possibility of employing serological tests targeting smooth *Brucella* species. As there is no standardised brucellosis testing scheme, no financial compensation and no compulsory policy measures for dogs in France exist, while systematic serological testing through the course of clinical symptoms cannot be imposed, especially when an isolated strain was already identified. This issue highlights the difficulty of collecting repeated and standardized samples in dogs.

Finding a cure for *Brucella* infected dogs remains a major issue due to the lack of an effective treatment [33]. Current options include neutering together with a long-term treatment with associations of antibiotics, including molecules that present potential toxic side-effects [47]. Although no guidelines are available for the treatment of *B. suis* infection in dogs, combination therapy with rifampicin and doxycycline would appear to be a possible effective therapy [19,46]. However, the use of rifampicin in dogs can instigate severe side-effects, such as vomiting, anorexia, lethargy and elevated levels of serum alanine aminotransferase (ALT; important transferase for liver functions) [48]. At the same time, rifampicin is guarded for treatment of human brucellosis and its use in animals may cause the development of antibiotic-resistant *B. suis* strains, threatening public health protection. Moreover, recurrence and/or persistence of the disease cannot be excluded, which leads us to consider a humane end to suffering as an alternative decision. Regarding the 13-year-old dog, neutering was performed before appearance of symptoms. It may be hypothesised that the dog presented a latent infection that reappeared with advanced age and/or a depressed immune system. Antibiotics were not sufficient to restore the overall health status of the dog.

The broad-spectrum wgSNP analysis, despite the large number of sequences, made it possible to locate these strains in a subclade including seven other French isolates (Figure 2), which were essentially isolated from wild animals (hares and wild boars but also from domestic pigs) between 1997 and 2022. This number increased to eight isolates through targeted analysis of BSB2 circulating in French territory (Figure 3), with cases mainly derived (seven out of eight strains) from infected pig herds isolated in France from 1996 to 2022. Although the 15 BSB2 French strains clustering with the two dog isolates (Figure 3) could not be related to a specific region, three distinct regions can be identified (the northwest, southwest and center of France; Figure 4). Interestingly, strains originating from the northwest and southwest regions of France were mainly isolated from outdoor breeding pigs, while isolates from central parts of country were found in wildlife (wild boars and hares) and pigs. The southwest region of France regroups many outdoor pig farms with local breeds, implying that pigs can be in direct contacts with wild animals. Based on the history of the strain isolated from the dog in this region (20-02069-2828), no livestock farms were found in the proximity of the dog’s residence, which reaffirms the wildlife origin hypothesis. The history of the second dog (22-3912-5948) is also not correlated with proximity to any type of pig breeding facility or direct contacts with wildlife or hunting. Assuming that porcine brucellosis outbreaks in France are related to brucellosis in wildlife, the main hypothesis seems to be an infection through contacts with body fluids and/or tissues from sylvatic reservoirs, such as hares and wild boars. Contacts may include sniffing, licking or eating these tissues. Currently, BSB2 is not considered as a pathogenic strain of *Brucella* and has never been isolated from dogs. In particular, in the case of the 13-year-old dog, *Brucella* infection could have occurred a long time before appearance of symptoms, and the clinical symptoms could emerge once animal started to suffer from other comorbidities. This possible latent infection complicates epidemiological investigations and the search for sources of contamination. Furthermore, the inability to determine with certainty the cause and time of infection may explain how this case occurred in an area with no previous cases of BSB2 in either pigs or wild animals.

Human cases of *B. suis* biovar 1, 3 and 5 have been reported all over the world [49,50,51,52]. So far, BSB2 strains are shown to be of low pathogenicity for humans, with only two cases reported in the literature until 2017, when in France, seven new cases were identified through national mandatory notification for brucellosis [12]. All patients were in contact with wild boars and their meat, and five of them also suffered from chronic medical conditions that could exacerbate disease or help *B. suis* infections. From all seven described cases, BSB2 strains were isolated. These cases also show that even with *Brucellae* of low pathogenic potential for humans, risks to public health augment with the increased numbers of infected animals and host species. These risks are even higher when we take into account that dogs are present in both rural and urban environments, their numbers are rising constantly, and they are in close contact with human populations of all ages and immune categories.

This study highlights both the role of wildlife in transmission of diseases such as brucellosis and the need to establish surveillance plans that include wildlife and environment, in order to better characterize the epidemiological situation and to limit risk of spill overs to domestic animals. Adopting a “One Health” approach would improve early detection of new outbreaks and minimize the risks of transmission to humans. This study also underlines the relevance of serological tests for classical smooth *Brucella* species in diagnoses of cases in dogs, since canine brucellosis is frequently described as concerning only rough *B. canis* strains.

## Figures and Tables

**Figure 1 pathogens-12-00792-f001:**
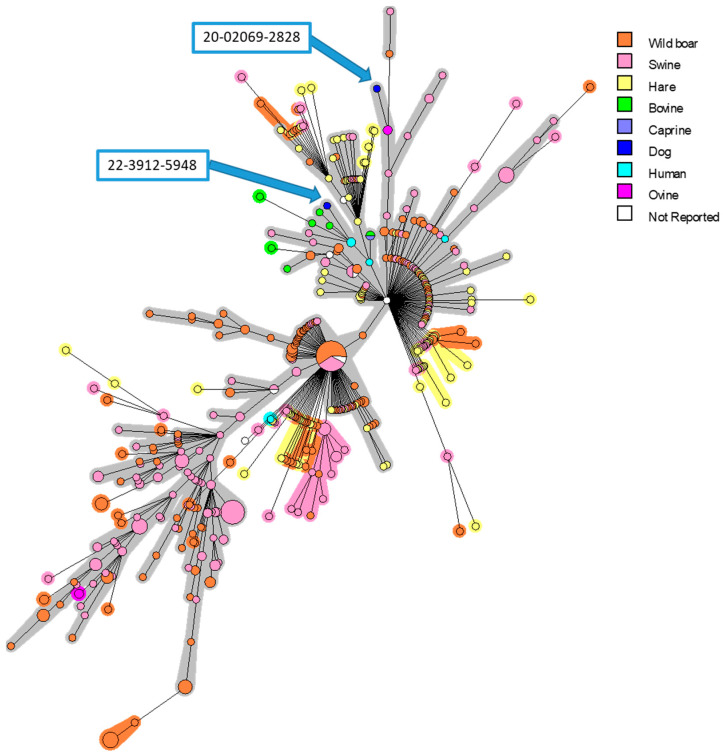
MLVA-16 minimum spanning tree describing relationships of 561 *B. suis* isolates originating from different European countries. Clustering and partitioning were generated with BioNumerics, using data as a character dataset with categorical distance coefficient and Minimum Spanning Tree method. Circles represent MLVA-16 genotypes, which are colored with respect to host-species, and size of circle indicates number of strains with that genotype. Two canine isolates presented in this work were marked with blue arrows.

**Figure 2 pathogens-12-00792-f002:**
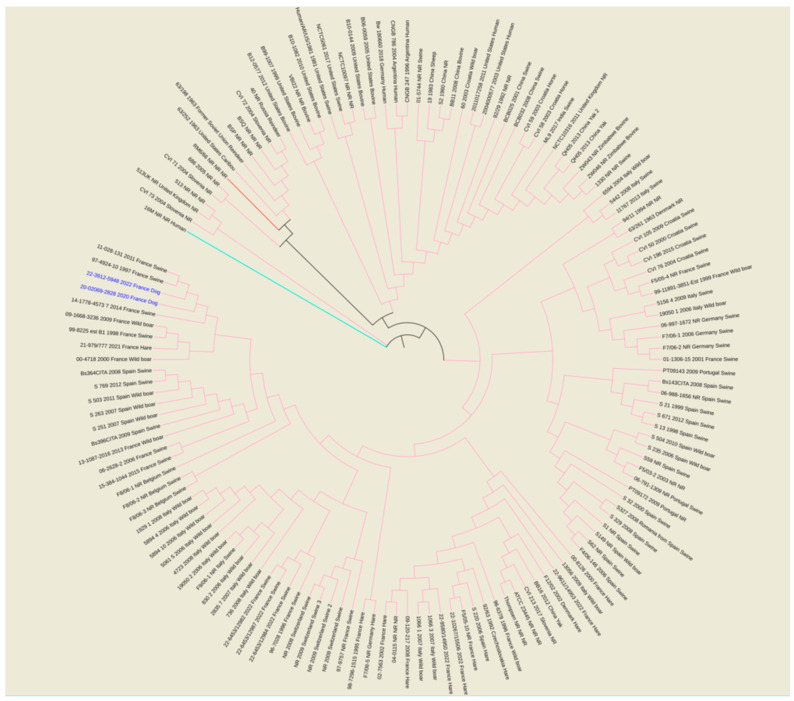
Maximum parsimony tree of *B. suis* isolates from various geographical regions of world (Africa, Asia, Europe, North America and South America). Parsimony tree was generated with BioNumerics using maximum parsimony algorithm from 8849 SNPs identified from complete genomes of 140 *B. suis* strains, using reference strain *B. melitensis* biovar 1 16 M and *B. canis* reference strains RM6/66, as two outliers. Phylogenetic tree was visualized with EMBL online toll “Interactive Tree of Life” (iTOL v6) and annotated with four concatenated datasets separated based on underscores (strain name, year of isolation, isolation country and host) colored in black and blue for all species and two isolates from dogs, respectively. Not reported data are marked with “NR”. All species are color-coded for branches: pink for *B. suis*, red for *B. canis* RM6/66 and magenta for *B. melitensis* 16M.

**Figure 3 pathogens-12-00792-f003:**
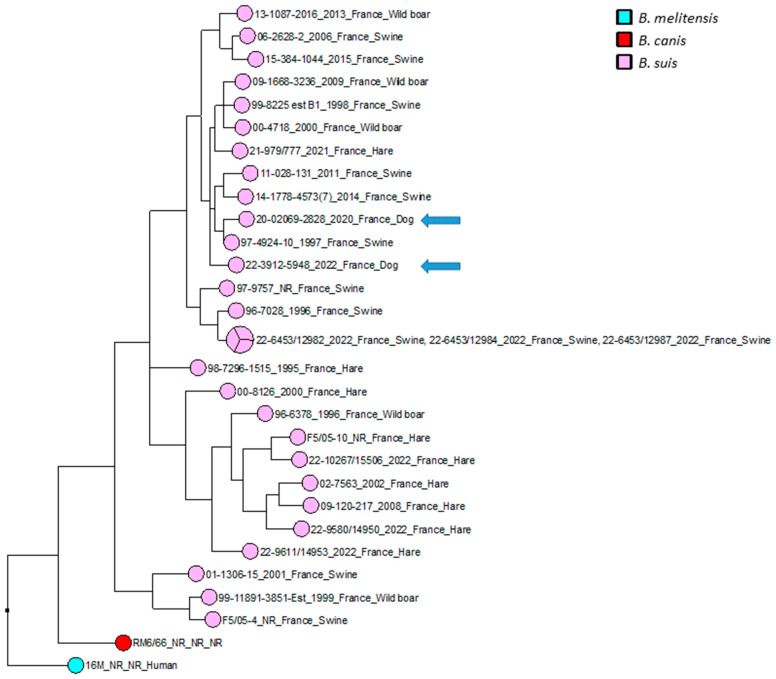
Maximum parsimony tree of *B. suis* biovar 2 isolates circulating in France. Parsimony tree was generated with BioNumerics using maximum parsimony algorithm from 9500 SNPs identified from complete genomes of 31 French *B. suis* strains, using reference strain *B. melitensis* biovar 1 16 M and *B. canis* reference strains RM6/66 as two outliers. Phylogenetic tree was annotated with four concatenated datasets separated based on underscores (strain name, year of isolation, isolation country and host). Not reported data are marked with “NR”. *B. suis* isolates are represented in pink, with two canine isolates represented by blue arrows. *B. canis* RM6/66 strain is represented in red, and referent *B. melitensis* 16M strain is represented in magenta.

**Figure 4 pathogens-12-00792-f004:**
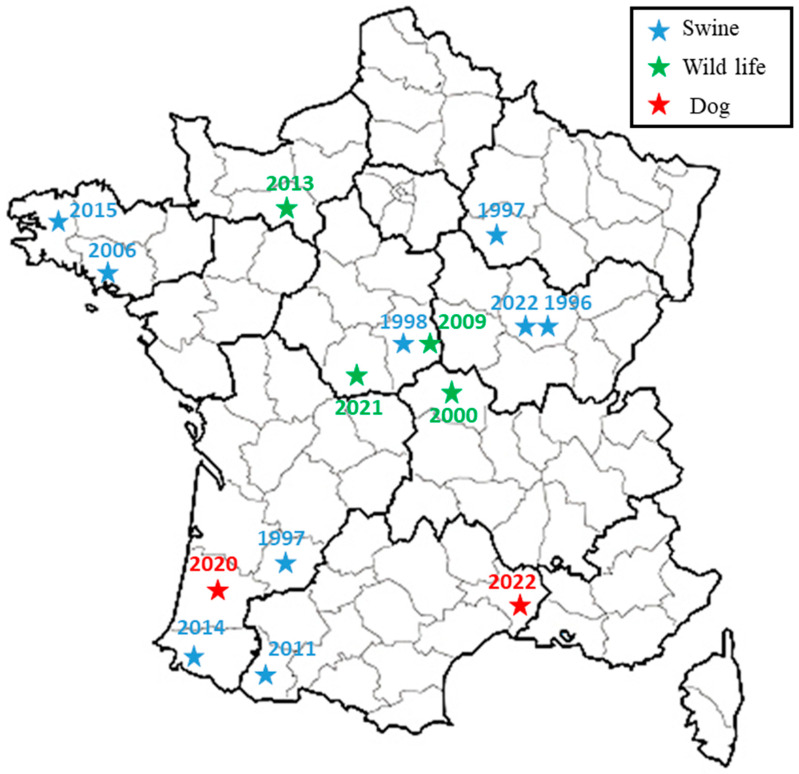
Spatial representation of French subclade related to two dog isolates. Previous isolations of BSB2 from wildlife (wild boars or hare) and pigs are presented with green and blue stars, respectively, while two canine isolates are presented with red stars.

**Table 1 pathogens-12-00792-t001:** Classical phenotypic characterization of two isolated strains (20-02069-2828, 22-03912-5948), compared to referent BSB2 and *B. canis* strains.

	*B. canis* RM6/66	*B. suis* bv 2 Thomsen	20-02069-2828	22-03912-5948
Morphology	R	S	S	S
CO_2_	-	-	-	-
H_2_S	-	-	-	-
Oxidase	+	+	+	+
Urease	+ ^a^	+ ^a^	+	+
A	-	+	+	+
M	-	-	-	-
R	+	-	-	-
Thionin	+	+	+	+
Fuchsin	(−)	-	-	-
Tb RTD	-	-	-	-
Tb 10^4^ RTD	-	+	+	+
Wb RTD	-	(+) ^b^	+	+
Iz RTD	-	(+) ^b^	PL	+
R/C RTD	+	-	-	-

R/S, colony morphology (rough/smooth), CO_2_ requirement, H_2_S production; agglutination with monospecific A, M and R (rough) antisera; dye (thionin and basic fuchsin) concentration 20 µg/mL in serum dextrose medium (1/50,000); + = growth or lysis by phages; - = no growth or lysis; PL, partial lysis; (+)/(−), most isolates positive/negative. ^a^ Rapid rate. ^b^ Some isolates are not or only partially lysed by phage.

## Data Availability

All data generated during this study were deposited in the European Nucleotide Archive (ENA) at EMBL-EBI under accession number PRJEB61443 (https://www.ebi.ac.uk/ena/browser/view/PRJEB61443 (accessed on 29 May 2023)).

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
