# Peer review of "Molecular Investigations of Two First Brucella suis Biovar 2 Infections Cases in French Dogs"

_pathogens, 2023, doi:10.3390/pathogens12060792_

Round 1

Reviewer 1 Report

Dear Authors,

The detection of B. suis infection in dogs is an interesting issue. In my opinion the paper is well written.

I have some suggestions, hoping to improve the quality of presentation.

Introduction: 

I suggest describing the monitoring programs going on in France (or Europe) for brucellosis. As far as I know, only B. abortus/melitensis are under constant and continuous surveillance, while no official data prevalence are available (apart from specific studies) for B. canis, B. ovis, B. suis etc. Therefore, reliable data on brucellosis prevalence in pigs, hares, dogs or wild boars are not available. This is also due to the imperfection and cross-reaction of serological diagnostic methods (cross-reaction among smooth Brucella species, obviously). If Authors do agree to this statement, I suggest adding this frame description.

Materials and Methods:

I suggest to better describe the sampling: the strain isolation procedure is described for organs but not for urine samples.

Serological methods are not described, but in the discussion (L. 296-297) a serological test for B. canis is cited. Why not to show this procedure?

At lines 296-297, I did not understand if the possibility to employ serological tests targeting smooth Brucella species was excluded because of lacking diagnostic test or because the hypothesis of this kind of infection was discarded.

I also suggest to introduce the two cases in the M&M section.

Results:

Lines 150-151: I suggest to better describe the clinical history that led to the decision of euthanize the dog. Was he suffering? Were the owners worried about the zoonotic risk? Why not to try another antibiotic treatment?

Lines 165-167: any follow up for the second dog? Did he survive/healed after neutering? Did he stop shedding Brucella?

Figure 1: the color legend is difficult to read, many colors are very similar

Discussion:

Line 263: I suggest changing to “In France, two dogs, respectively 13 and 5 years old”

The discussion do not consider B. ovis: is it not present in France?

I suggest considering the diagnostic limits of serology in the hypothesis of monitoring/surveillance programs.

The passive surveillance and the direct diagnosis on symptomatic dogs is important, but to investigate the real prevalence of the infection, serological screening should be considered, taking into account their important limits in sensitivity and specificity.

Author Response

Dear Reviewers,

The authors are grateful to the reviewers for their critical evaluation, helpful suggestions and corrections. We have made changes according to their constructive comments and introduced modifications to the manuscript to clarify our work. All issues raised by the reviewers have been addressed here below. The corrections are visible in the revised manuscript (deleted are marked with yellow and added text in blue).

If you consider that additional changes would be needed, we rest at your disposal.

All the best!

Luca Freddi

Comments and Suggestions for Authors from Reviewer 1 :
Dear Authors,
The detection of B. suis infection in dogs is an interesting issue. In my opinion the paper is well written.
I have some suggestions, hoping to improve the quality of presentation.

Introduction:
I suggest describing the monitoring programs going on in France (or Europe) for brucellosis. As far as I know, only B. abortus/melitensis are under constant and continuous surveillance, while no official data prevalence are available (apart from specific studies) for B. canis, B. ovis, B. suis etc. Therefore, reliable data on brucellosis prevalence in pigs, hares, dogs or wild boars are not available. This is also due to the imperfection and cross-reaction of serological diagnostic methods (cross-reaction among smooth Brucella species, obviously). If Authors do agree to this statement, I suggest adding this frame description.

Changes were made from line 29 to 37: In France, like in other EU countries, the prevention, control and eradication of brucellosis is regulated by Animal Health Law (AHL; regulation EU No 2018/1882 ; EUR-Lex - 32018R1882 - EN - EUR-Lex (europa.eu)). The AHL includes the surveillance and management of B. abortus, B. melitensis and B. suis in production animals and wildlife. However, the AHL does only consider dogs as potential reservoirs for B. abortus, B. melitensis and B. suis. In France, surveillance is organised with departmental veterinary diagnostic laboratories which perform first-line analyses. When the Brucella suspected colonies are detected, the isolated strains are sent to National Referent Laboratory (LNR) for animal brucellosis, for the genetic confirmation and further species characterisation.

Materials and Methods:
I suggest to better describe the sampling: the strain isolation procedure is described for organs but not for urine samples.

According to the reviewer’s comments, the section 2.1 “Bacterial cultivation and strains isolation” in materials and methods has been revised including detail for bacteriological cultures: All collected samples for bacteriological analyses, were analyzed using routine bacteriological procedures in the local veterinary diagnostic laboratories, in accordance with the safety regulation in force. Briefly, the urine samples collected from in the 13-years old Border collie (first reported case), were directly cultivated on Chocolate PolyViteX (BioMerieux, France) and chromogenic CPS media (BioMerieux, France). On the other hand, tissue biopsies of the scrotal region collected from the 5-years old German shepherd (second reported case), were cut with a surgical single use scalpel, and then 10 g was homogenised and diluted in 1/2 to 1/5 ratio in phosphate buffered saline (PBS) solution (0.9%). The homogenate was then plated onto Chocolate PolyViteX and Columbia agar (Sigma-Aldrich, Germany) with sheep’s blood (5%) media. The resulting colonies of the two cases were purified by replating, and then analyzed by MALDI-TOF for the first-line identification purposes. The two isolates (20-02069-2828 and 22-03912-5948) suspected to be Brucella, were transferred to the French National Reference Laboratory for animal Brucellosis (ANSES, Maisons-Alfort) to confirm the Brucella genus and determine the species, in accordance with the safety regulation in force.

Serological methods are not described, but in the discussion (L. 296-297) a serological test for B. canis is cited. Why not to show this procedure?

We thank reviewer for noticing this. At first, based on the anamnesis in both cases brucellosis was not included in differential diagnosis and therefore no serological test was performed neither in departmental diagnostic laboratory nor in our National Referent Laboratory for animal brucellosis, which was contacted only after isolation of the strains (according to national procedure). In the first case of 13-years old dog the health status deteriorated rapidly and during the bacteriological identification and genetic analyses, the owners decided to euthanize it, leaving us without the possibility of further sampling. In the second case the only serological analysis was performed by departmental laboratory targeting only rough Brucella canis, which rendered negative results. Additionally, our laboratory cannot impose further sampling, especially when an isolated strain has been already identified (Line 338-342). This limit was explained in the discussion section of the article: “Because there is no standardised brucellosis testing scheme, no financial compensation and no compulsory policy measures for dogs in France, systematic serological testing through the course of clinical symptoms cannot be imposed, especially when an isolated strain has been already identified. This highlights the difficulty to collect repeated and standardized samples in dogs. “

At lines 296-297, I did not understand if the possibility to employ serological tests targeting smooth Brucella species was excluded because of lacking diagnostic test or because the hypothesis of this kind of infection was discarded.

The text has been corrected as serological test has been performed only for the second case. The only one rapid slide agglutination tests (RSAT) test targeting rough Brucella strains (as B. canis) has been performed for a second case. No other serological tests targeting smooth Brucella strains were performed, not for absence of available test, rather failure of veterinary prescription highlights the lack of knowledge of canine brucellosis in France (not considering
relevant the potential infections in dogs by B. suis, B. abortus or B. melitensis). This issue is mainly related to the absence of specified policy on control of suspected and non-suspected dogs. As previously mentioned, our laboratory cannot impose additional samples for serological tests, especially when an isolated strain has been already identified.

I also suggest to introduce the two cases in the M&M section.

According to the reviewer’s comments, the section 2.1 “Bacterial cultivation and strains isolation” in materials and methods has been revised. However, the authors prefer not to detail the two cases in order to avoid repetition provided in section 3.1.

Results:

Lines 150-151: I suggest to better describe the clinical history that led to the decision of euthanize the dog. Was he suffering? Were the owners worried about the zoonotic risk? Why not to try another antibiotic treatment?

The reasons leading up to the decision to put the dog to sleep were explained following reviewer suggestion (Line 182-183): The dog was treated with Doxycycline for 20 days, without recovery, which led to decision to put it to sleep considering the dog's age and deteriorated health condition.

Lines 165-167: any follow up for the second dog? Did he survive/healed after neutering? Did he stop shedding Brucella?

In the second case the only serological analysis was performed by departmental laboratory targeting only rough Brucella canis, which rendered negative results. Because there is no specific brucellosis regulation for dogs in France, we cannot impose the regular serological testing through the course of clinical symptoms or after, when animal was cured. Although dog survived we have no legal means to impose follow up. As the dog did not present any symptoms following the surgery and antibiotics treatment, the owner did not want to continue the follow-up. According to the reviewer’s comment, the following sentence has been added (Line 202-203): “After clinical signs retracted, no additional sampling was performed to follow up on the presence of specific BSB2 antibodies, nor shedding of live bacteria.”

Figure 1: the color legend is difficult to read, many colors are very similar

We agree with the reviewer and the figure has been revised using the most visible colours among those available in the management software: orange for wild boar, pink for swine, yellow for hare, green for bovine, violet for caprine, blue for dog, magenta for human, fuchsia for ovine and white for not reported information.

Discussion:
Line 263: I suggest changing to “In France, two dogs, respectively 13 and 5 years old”

Corrected.

The discussion do not consider B. ovis: is it not present in France?

We thank reviewer for this remark. In France Rev.1 vaccination has been prolonged up to 2017 in endemic regions for B. ovis together with a surveillance plan based on serological tests. Therefore, the last B. ovis field strain was isolated long time ago. Moreover, B. ovis has a quite limited host range compared to other Brucella species, which does not include Carnivora. Therefore we have not considered this species as a potential risk for dogs.

I suggest considering the diagnostic limits of serology in the hypothesis of monitoring/surveillance programs.

We agree with a reviewer and this paper goes to show that serological surveillance programs are needed to implement in France and EU in general, as has been mentioned in our conclusions. All results reported in this paper are derived from direct diagnostics and characterisation, and we don’t have large enough pool of data to discuss diagnostic limits of serology in dogs. However, for now there is no specific brucellosis regulation for dogs, and we cannot impose the regular or additional sampling for serological testing, when an isolated strain has been already identified. This limit was explained in the discussion section of the article (Line 338-342): “Because there is no standardised brucellosis testing scheme, no financial compensation and no compulsory policy measures for dogs in France, systematic serological testing through the course of clinical symptoms cannot be imposed, especially when an isolated strain has been already identified. This highlights the difficulty to collect repeated and standardized samples in dogs. “

The passive surveillance and the direct diagnosis on symptomatic dogs is important, but to investigate the real prevalence of the infection, serological screening should be considered, taking into account their important limits in sensitivity and specificity.

We agree with this comment. However, for now there is no specific brucellosis regulation for dogs, and we cannot impose the regular serological testing, especially without sufficient evidence that dogs are Brucella sp. reservoirs because then the cost of such testing would fall on owners and not French government.

Reviewer 2 Report

1. This study is interesting. B. suis bv.2 strains from dogs were identified and analyzed by MLVA and wgSNP to find the infection sources. Seroprevelance of dogs in France is very important to evaluate the public health risk.  The releated information should be included. 

2. A recent seroprevalence study, for example, found that nearly one in 10 (9.3%) dogs in eastern Australia, exposed to feral pigs or their meat products, are seropositive for B. suis [50,51].  How to discriminate B. suis infection from serum tests? 

Fine.

Author Response

Dear Reviewers,

The authors are grateful to the reviewers for their critical evaluation, helpful suggestions and corrections. We have made changes according to their constructive comments and introduced modifications to the manuscript to clarify our work. All issues raised by the reviewers have been addressed here below. The corrections are visible in the revised manuscript (deleted are marked with yellow and added text in blue).

If you consider that additional changes would be needed, we rest at your disposal.

All the best!

Luca Freddi

Comments and Suggestions for Authors from Reviewer 2 :

  1. This study is interesting. B. suis bv.2 strains from dogs were identified and analyzed by MLVA and wgSNP to find the infection sources. Seroprevelance of dogs in France is very important to evaluate the public health risk.  The releated information should be included. 

We agree with the reviewer regarding the importance to know the brucellosis seroprevalence in French dogs. However, to date there has been no systematic or cross-sectional study performed in any European country to evaluate the prevalence of canine brucellosis (including rough and/or smooth Brucella strains). To better specify that from line 85 to 88, sentence has been modified to: However, no study has systematically analysed the circulation of B. suis infection in dogs (including serological survey in European countries), suggesting that canine brucellosis due to this Brucella species is potentially under-reported, probably due to the limitations of accurate diagnosis, lack of regulation and the low probability of human infection [24,33].

  1. A recent seroprevalence study, for example, found that nearly one in 10 (9.3%) dogs in eastern Australia, exposed to feral pigs or their meat products, are seropositive for B. suis [50,51].  How to discriminate B. suis infection from serum tests? 

We agree with the reviewer that serum tests cannot discriminate Brucella species infection. Nevertheless, we merely reported the peer reviewed publication on occurrence of B. suis in dongs and authors’ conclusions (Kneipp et al. 2021, doi: https://doi.org/10.3389/fvets.2021.727641). According to reviewer remark, the phrase has been modified to (Line 322-325): A recent seroprevalence study, for example, found that nearly one in 10 (9.3%) dogs in eastern Australia, exposed to feral pigs or their meat products, are seropositive to smooth Brucella spp. (including B. suis) [50,51].

Reviewer 3 Report

The case report is very interesting.

Some suggestions to improve the manuscript

Line 38: one epidemiological study etc; do you mean causing brucellosis? Please correct ( explain this sentence  )

Line 43-44: please rewrite this sentence

Line 78: a brief comment about the  aim of this study should be good.

MM: I don’t understand the structure, the separation between MM and results, could you firstly introduce the case ( lines 133 Etc) and then the isolation ( lines 80 etc). ?

Lines 240-251: this part introduces Brucella species in general, should be moved to the introduction . please correct and write a short sentence for Discussion.

Lines 276-278: this sentence is quite confusing for the reader.  Dogs are considered dead end hosts of smooth brucellae  such as abortus ( melitensis) , while B. canis is host adapted, please rewrite this sentence

Line 297: shown

none

Author Response

Dear Reviewers,

The authors are grateful to the reviewers for their critical evaluation, helpful suggestions and corrections. We have made changes according to their constructive comments and introduced modifications to the manuscript to clarify our work. All issues raised by the reviewers have been addressed here below. The corrections are visible in the revised manuscript (deleted are marked with yellow and added text in blue).

If you consider that additional changes would be needed, we rest at your disposal.

All the best!

Luca Freddi

Comments and Suggestions for Authors from Reviewer 3:

The case report is very interesting.

Some suggestions to improve the manuscript

- Line 38: one epidemiological study etc; do you mean causing brucellosis? Please correct ( explain this sentence  )

The text has been corrected following reviewer suggestion (Line 48-51): One epidemiological study of BSB2 causing brucellosis in French pork production, identified that the likelihood of transmission of infection among free-range farms is low due to lack of commercial exchange of animals [5], suggesting random contacts with infected wildlife as a major source of infection.

- Line 43-44: please rewrite this sentence

Done (Line 53-55): Since 1993, between 20% and 35% positive Brucella serological reactions (Rose Bengal Test and/or Complement Fixation Test) were observed in hunted or found dead wild boars in several departments throughout the country [5].

- Line 78: a brief comment about the aim of this study should be good.

According to reviewer remark, the following phrase has been added to the line 89-92: Our results demonstrate, for the first time, BSB2 infection of exposed dogs, highlighting the high survival rate of BSB2 in the environment, potential to infected canids, and how even domestic pets without evident contact with wild animals or pig industries could be source for human infections.

- MM: I don’t understand the structure, the separation between MM and results, could you firstly introduce the case ( lines 133 Etc) and then the isolation ( lines 80 etc). ?

In accordance with Pathogens template, the authors decided to include the “Presentation of cases” in the result section and not in Materials and methods. In section 2.1 (Bacterial cultivation and strains isolation) we briefly described only the techniques used for strain isolation.

- Lines 240-251: this part introduces Brucella species in general, should be moved to the introduction. please correct and write a short sentence for Discussion.

The introduction to the Discussion section has been shortened as reviewer suggested (Line 279-292): Brucellosis is a highly contagious bacterial zoonosis mainly caused by ingestion of contaminated unpasteurized milk, dairy produce, not enough thermically treated meat, or close contact with secretions of infected animals (semen, aborted or parturition tissues and fluids). Etiological agent of brucellosis are bacteria from Brucella genus, with 12 species currently described. Considering that the whole genus shares ~97% of genetic similarity [46], it is unclear how host preferences are developed. However, the plasticity of the genus continues to be discovered.

- Lines 276-278: this sentence is quite confusing for the reader.  Dogs are considered dead end hosts of smooth brucellae such as abortus (melitensis), while B. canis is host adapted, please rewrite this sentence

Sentence changed to (Line 315-317): At the same time, canine brucellosis in Europe has been reported only sporadically, related mainly to B. melitensis, B abortus and since very recently to B. canis.

- Line 297: shown

Corrected.

Reviewer 4 Report

I don’t really understand what happened with this story. The characterisation of the isolates and the phylogenetics look impressive – but I am a clinician and so i cannot comment further on that work. Its interesting it was B suis serotype 2.

Here is what is written:

In April 2020, a 13-year-old male neutered Border collie presented with thickening of 133 the bladder, urethral duct and prostate with pain related to prostatic palpation was 134 brought to a veterinary clinic. Urine was cloudy, and all other clinical signs were associ-135 ated with cystitis and/or prostatitis, suggesting an infection of the urogenital tract. The 136 animal presented no evident signs of brucellosis. This dog was acquired through e-com-137 merce at the age of 4 years and originated from South West France (Pyrennées Atlantiques 138 department). Since then, the dog lived in the Landes department (also South West) as only 139 pet in the household, and was regularly followed by a veterinarian. The dog used to take 140 long walks in the surrounding forests where it could have been in contact with other do-141 mestic and/or wild animals and their excrements. In order to exclude bacterial infection 142 and/or prostatic cancer, the dog was examined in the local veterinary clinic where cyto-143 logical and bacteriological analyses of urine were performed. The ultrasound analysis did 144 not show any evidence of tumoral tissues. The cytological results highlighted high level 145 of leucocytes, confirming an ongoing infection of the urogenital tract, otherwise negative 146 results were obtained from urine bacteriology. Considering the persistence of clinical 147 Pathogens 2023, 12, x FOR PEER REVIEW 4 of 14 signs, a new urine sample was obtained 7 days apart, together with ultrasound. The urine 148 culture was found to be positive for the Brucella genus with an amount of 104 colony form-149 ing units (CFU) after three days of incubation, confirming the ongoing infection. The dog 150 was treated with Doxycycline for 20 days without recovery, which led to decision to put 151 it to sleep. 152

Another case of an infected dog occurred in February 2022. The 5-years-old male 153 German shepherd presented high fever with an abnormal enlargement of the genital ap-154 paratus suggesting a bilateral orchitis. The dog was born in the Gard department (South 155 of France), where it was adopted by a family living in the countryside of Beaucaire, an 156 area where wild boars are largely prevalent. As first complementary investigations, a 157 blood analysis was performed, showing leucocytosis and lymphopenia, indicating an on-158 going infection. Since the dog had no historical evidence of brucellosis, the local veterinary 159 clinic decided to treat it initially with combination of amoxicillin, clavulanic acid and a 160 non-steroidal anti-inflammatory drug. As there was no improvement, instead a deterio-161 ration of the reproductive organs detected by echography, a neutering with scrotectomy 162 was performed to avoid any potential risk of spread and to identify the cause of the infec-163 tion. Following surgery, treatment was strengthened with the addition of second line-an-164 tibiotic. The retrieved testicles were sent to a diagnostic laboratory to perform a bacterial 165 culture, the result of which showed the presence of colonies belonging to the genus Bru-166 cella.

I have no understanding of what really happened. I can understand samples from the dogs being collected to characterise if urosepsis was present – but why didn’t the samples go to a ROUTINE vet lab, how did it end up in a BSL-3 facility?? Colonies of Brucella suis are not characteristic – they are very on-reactive and do not look like the standard bacteria we isolate from the urogenital tract of dogs. In Australia – we have a high index of suspension if the dog was used for pig hunting or ate  feral pig meat, and usually its orchitis or epididymitis that raises the spectre of brucellosis. Its almost unheard of for a male dog that has been castrated to develop Brucella in the prostate (vertebral osteomyelitis is much more common when they have been desexed). If you send a sample to a normal vet lab in 2023 – usually they make an ID by sticking a colony on the MADLI-tof machine – only you are not suppose to do that with this organism because you only play with it  in a secure incubator. Without first doing a Rose Bengal test – you have no reason to think it will be Brucella – so normally no special precautions are made – and its examined on the open bench – which then means all the people in that lab seen to go on preventive drugs (doxycycline and rifampicin) for a month. WHAT REALLY HAPPENED HERE?

The first dog was put to sleep. The 2nd dog was castrated and treated with antibiotics – the details of which are not provided. WHY NOT?

In terms of the discussion, its really important to state that Brucella suis is a eminently treatable disease in digs – most can be easily cured by castration combined with rifampicin and doxycycline – which are cheap antibiotics. There is no place for euthanasia. Enormous work has shown that dogs are almost always infectious for people.

In this case – it is REMARKABLE that bacteria were present in the urine. Can we seethe images of the prostate? In Australia – large number of seropositive cases have been subjected to urine culture and PCR and none were positive (very rarely semen is positive). So the dog in this report is UNUSUAL – but there is so little clinical data, we cannot even guess if the focus in the infection was in the clinic or the prostate.

The authors need to REVIEW all the clinical data, read all the recent papers from Australia, and rewrite the sections on prognosis and therapy. WE also need to know is they were positive using serology – Rose Bengal, ELISA, complement fixation.

The whole clinical section needs to be rewritten, more data supplied, and serological tests performed using stored serum samples. We also need long term follow up for case 2. 

Because the authors are FRench some expressions are clumsy - a subeditor needs to tidy this up - its minor

Author Response

Dear Reviewers,

The authors are grateful to the reviewers for their critical evaluation, helpful suggestions and corrections. We have made changes according to their constructive comments and introduced modifications to the manuscript to clarify our work. All issues raised by the reviewers have been addressed here below. The corrections are visible in the revised manuscript (deleted are marked with yellow and added text in blue).

If you consider that additional changes would be needed, we rest at your disposal.

All the best!

Luca Freddi

Comments and Suggestions for Authors from Reviewer 4 :

I don’t really understand what happened with this story. The characterisation of the isolates and the phylogenetics look impressive – but I am a clinician and so i cannot comment further on that work. Its interesting it was B suis serotype 2.

Here is what is written:

In April 2020, a 13-year-old male neutered Border collie presented with thickening of 133 the bladder, urethral duct and prostate with pain related to prostatic palpation was 134 brought to a veterinary clinic. Urine was cloudy, and all other clinical signs were associ-135 ated with cystitis and/or prostatitis, suggesting an infection of the urogenital tract. The 136 animal presented no evident signs of brucellosis. This dog was acquired through e-com-137 merce at the age of 4 years and originated from South West France (Pyrennées Atlantiques 138 department). Since then, the dog lived in the Landes department (also South West) as only 139 pet in the household, and was regularly followed by a veterinarian. The dog used to take 140 long walks in the surrounding forests where it could have been in contact with other do-141 mestic and/or wild animals and their excrements. In order to exclude bacterial infection 142 and/or prostatic cancer, the dog was examined in the local veterinary clinic where cyto-143 logical and bacteriological analyses of urine were performed. The ultrasound analysis did 144 not show any evidence of tumoral tissues. The cytological results highlighted high level 145 of leucocytes, confirming an ongoing infection of the urogenital tract, otherwise negative 146 results were obtained from urine bacteriology. Considering the persistence of clinical 147 Pathogens 2023, 12, x FOR PEER REVIEW 4 of 14 signs, a new urine sample was obtained 7 days apart, together with ultrasound. The urine 148 culture was found to be positive for the Brucella genus with an amount of 104 colony form-149 ing units (CFU) after three days of incubation, confirming the ongoing infection. The dog 150 was treated with Doxycycline for 20 days without recovery, which led to decision to put 151 it to sleep. 152

Another case of an infected dog occurred in February 2022. The 5-years-old male 153 German shepherd presented high fever with an abnormal enlargement of the genital ap-154 paratus suggesting a bilateral orchitis. The dog was born in the Gard department (South 155 of France), where it was adopted by a family living in the countryside of Beaucaire, an 156 area where wild boars are largely prevalent. As first complementary investigations, a 157 blood analysis was performed, showing leucocytosis and lymphopenia, indicating an on-158 going infection. Since the dog had no historical evidence of brucellosis, the local veterinary 159 clinic decided to treat it initially with combination of amoxicillin, clavulanic acid and a 160 non-steroidal anti-inflammatory drug. As there was no improvement, instead a deterio-161 ration of the reproductive organs detected by echography, a neutering with scrotectomy 162 was performed to avoid any potential risk of spread and to identify the cause of the infec-163 tion. Following surgery, treatment was strengthened with the addition of second line-an-164 tibiotic. The retrieved testicles were sent to a diagnostic laboratory to perform a bacterial 165 culture, the result of which showed the presence of colonies belonging to the genus Bru-166 cella.

I have no understanding of what really happened. I can understand samples from the dogs being collected to characterise if urosepsis was present – but why didn’t the samples go to a ROUTINE vet lab, how did it end up in a BSL-3 facility?? Colonies of Brucella suis are not characteristic – they are very on-reactive and do not look like the standard bacteria we isolate from the urogenital tract of dogs. In Australia – we have a high index of suspension if the dog was used for pig hunting or ate  feral pig meat, and usually its orchitis or epididymitis that raises the spectre of brucellosis. Its almost unheard of for a male dog that has been castrated to develop Brucella in the prostate (vertebral osteomyelitis is much more common when they have been desexed). If you send a sample to a normal vet lab in 2023 – usually they make an ID by sticking a colony on the MADLI-tof machine – only you are not suppose to do that with this organism because you only play with it  in a secure incubator. Without first doing a Rose Bengal test – you have no reason to think it will be Brucella – so normally no special precautions are made – and its examined on the open bench – which then means all the people in that lab seen to go on preventive drugs (doxycycline and rifampicin) for a month. WHAT REALLY HAPPENED HERE?

We thank reviewer for his comments. The aim of this study, was to characterize the two B. suis bv2 strains isolated from dogs and to detect the potential sources of infection through phylogenetic investigations (MLV-16 and wgSNPs). The clinical cases were described succinctly in the first part of the Results section without carefully describing the entire clinical analysis process. In France, samples are at first sent to a local vet laboratory. However, identification of Brucella species from animal samples is confirmed only by the National Reference Laboratory (LNR). Since neither of two dogs had history related to brucellosis, and no anamnestic data indicated Brucella for differential diagnosis, the clinical samples were send to local diagnostic laboratories to perform a routine bacterial culture analyses. As mentioned by the reviewer, laboratories performed the routine MALDI-TOF analysis in accordance with the safety regulation in force, which isolated the strains suspected to belong to the genus Brucella. However, there is currently more and more evidence that MALDI-TOF spectral libraries are not complete and cannot distinguish between the Brucella genus and Ochrobactrum, nor can perform species identification (precise only up to the genus level). Hence, to respect the French regulation, suspected colonies (after a first MALDI-TOF based identification) were directly transferred to the LNR’s BSL-3 facility to confirm the Brucella genus identification and determine the species. Once the Brucella genus was confirmed by the LNR, the competent authorities were informed in order to assess and managed any possible risks of human infection.

To clarify performed analyses, treatments and outcomes, we detailed further description of two cases.

The first dog was put to sleep. The 2nd dog was castrated and treated with antibiotics – the details of which are not provided. WHY NOT?

We thank reviewer for this comments. We now added that dog survived, but current French laws prevent us to impose further sampling and analyses on dogs (also accentuated in the paper) (Line 201-203): After clinical signs retracted, no additional sampling was performed to follow up on the presence of specific BSB2 antibodies, nor shedding of live bacteria.

In terms of the discussion, its really important to state that Brucella suis is a eminently treatable disease in digs – most can be easily cured by castration combined with rifampicin and doxycycline – which are cheap antibiotics. There is no place for euthanasia. Enormous work has shown that dogs are almost always infectious for people.

We thank reviewer for this comment. The authors agree that antibiotic therapy (mostly rifampicin and doxycycline) following sterilization is available treatment for B. suis infected dogs which would reduce the risks of disease spreading. However, the scientific evidence unequivocally considers also euthanasia as CURRENTLY the only method to guarantee reducing the public health risks to zero (as reported by James et al. 2017, https://doi.org/10.1111/avj.12550). In addition, adopting a no-kill strategy with the employment of antibiotics therapy may carry the risk of potential toxic side effects occurring in the individual animal. Unfortunately, there are currently no commonly accepted criteria to support the decision of therapeutic treatment versus euthanasia, given the absence of universally accepted treatment protocols for B. suis-infected animals in Europe, which leaves the choice of antimicrobials and the duration of treatment a decision for veterinary clinicians.

In this case – it is REMARKABLE that bacteria were present in the urine. Can we see the images of the prostate? In Australia – large number of seropositive cases have been subjected to urine culture and PCR and none were positive (very rarely semen is positive). So the dog in this report is UNUSUAL – but there is so little clinical data, we cannot even guess if the focus in the infection was in the clinic or the prostate.

We thank reviewer for this comment. The ultrasound images of testis can be available upon specific individual request. Since this is not the characteristic clinical case report, authors considered that these images do not contribute to genetic analyses presented in this paper.

The authors need to REVIEW all the clinical data, read all the recent papers from Australia, and rewrite the sections on prognosis and therapy. WE also need to know is they were positive using serology – Rose Bengal, ELISA, complement fixation.

The whole clinical section needs to be rewritten, more data supplied, and serological tests performed using stored serum samples. We also need long term follow up for case 2. 

We thank reviewer for this comment. Additional clinical and diagnostic information have been provided in response to previous comments of all four reviewers.

Round 2

Reviewer 2 Report

The authors reply and modidy all comments.

Author Response

We thank reviewer for his/her suggestions.

Reviewer 4 Report

The paper is much improved for the revision. The References are DOUBLE numbered. The use of scientific terms all thru the references is INCORRECT – e.g. instead of Brucella suis, they have Brucella Suis.

The microbiology and molecular biology in this paper is  excellent. The veterinary science is poor. Brucella suis infections are readily cured by  castration or ovariohysterectomy combined with combination antimicrobial therapy with rifampicin and  doxycycline, with the occasional case also needing gentamicin. These papers are easy to find. Surely the vets can GOOGLE  a key word search to find the papers when confronted with an unusual infection, and give the dogs the benefit of appropriate therapy. This needs to be emphasised in the Discussion. The cure of the 2nd case despite  choosing a less effective treatment regimen shows this is not a difficult end to achieve. Note the 2 recent papers by Cathy Kneipp which have not been cited – one in Aust Vet Journal, and the other in J Vet Internal medicine. It pays to GOOGLE recent papers before completing revision so this paper can be as up to date as possible

MY wekfare concerns relate to the fact that with  well planned treatment this is a treatable disease - and the authors need to explain which drug combinations are most effective  for therapy

Author Response

We thank reviewer for his/her comments. The References section has been revised in order to eliminate the double numbers, correct the scientific terms and add the missed paper by Kneipp et al. 2023. Considering the antibiotic treatment, the authors cannot recommend which molecules can be considered more effective, since no accepted treatment guidelines are available for B. suis infection in dogs at present (this is also mentioned by Kneipp et al. 2023; Doi: 10.1111/jvim.16678). Moreover, our laboratory is in charge of detection of Brucella in monitored species, but cannot enforce treatments, outside of accepted protocols, for the clinical treatment of any infected animal species. However, we agree that the therapy options should be discussed, so the possible therapy with rifampicin and doxycycline has been mentioned (Line 346-348). On the other hand, the use of rifampicin, which is guarded specifically for human treatments of heavily resistant bacteria, has already been proven to be able to produce significant side effects in dogs (Bajwa et al. 2013, doi: https://doi.org/10.1111/vde.12083). In the same time, the use of rifampicin in dogs raises the possibility of antimicrobial resistance in B. suis creating even bigger problems for public health protection. These issues have been added to Discussion (Line 348-353).